# Oxidative Stress-Induced Pentraxin 3 Expression Human Retinal Pigment Epithelial Cells Is Involved in the Pathogenesis of Age-Related Macular Degeneration

**DOI:** 10.3390/ijms20236028

**Published:** 2019-11-29

**Authors:** Narae Hwang, Min-Young Kwon, Je Moon Woo, Su Wol Chung

**Affiliations:** 1School of Biological Sciences, College of Natural Sciences, University of Ulsan, 93 Daehak-ro, Ulsan 44610, Korea; skfo1319@naver.com (N.H.); youngpower-v@hanmail.net (M.-Y.K.); 2Division of Pulmonary and Critical Care Medicine, Brigham and Women’s Hospital and Harvard Medical School, Boston, MA 02115, USA; 3Department of Ophthalmology, Ulsan University Hospital, University of Ulsan College of Medicine, Ulsan 44033, Korea; limbus68@naver.com

**Keywords:** pentraxin 3, human retinal pigment epithelial cells, age-related macular degeneration, oxidative stress, sodium iodate antioxidants

## Abstract

(1) Background: Age-related macular degeneration (AMD) is closely related with retinal pigment epithelial (RPE) cell dysfunction. Although the exact pathogenesis of AMD remains largely unknown, oxidative stress-induced RPE damage is believed to be one of the primary causes. We investigated the molecular mechanisms of pentraxin 3 (PTX3) expression and its biological functions during oxidative injury. (2) Methods: Using enzyme-linked immunosorbent assays and real-time reverse transcription-polymerase chain reaction, we analyzed mRNA and protein levels of PTX3 in the presence or absence of oxidative stress inducer, sodium iodate (NaIO_3_), in primary human H-RPE and ARPE-19 cells. Furthermore, we assessed cell death, antioxidant enzyme expression, and AMD-associated gene expression to determine the biological functions of PTX3 under oxidative stress. (3) Results: NaIO_3_ increased PTX3 expression, in a dose- and time-dependent manner, in H-RPE and ARPE-19 cells. We found phosphorylated Akt, a downstream target of the PI3 kinase pathway, phosphor- mitogen-activated protein kinase kinase 1/2 (ERK), and intracellular reactive oxygen species (ROS) were predominantly induced by NaIO_3_. NaIO_3_-induced PTX3 expression was decreased in the presence of phosphoinositide 3 (PI3) kinase inhibitors, ERK inhibitors, and ROS scavengers. Furthermore, NaIO_3_ enhanced mRNA expression of antioxidant enzymes such as glucose-6-phosphate dehydrogenase (*G6PDH*), *catalase (CAT)*, and glutathione S-reductase (*GSR*) in the control shRNA expressing RPE cells, but not in hPTX3 shRNA expressing RPE cells. Interestingly, NaIO_3_ did not induce mRNA expression of AMD marker genes, such as *complement factor I* (*CFI*), *complement factor H* (*CFH*), *apolipoprotein E* (*APOE*), and *toll-like receptor 4* (*TLR4*) in hPTX3 shRNA expressing RPE cells. 4) Conclusions: These results suggest that PTX3 accelerates RPE cell death and might be involved in AMD development in the presence of oxidative stress.

## 1. Introduction

The retinal pigment epithelium (RPE), a monolayer located between the photoreceptors and the choroid, is essential for survival of the retina, including maintaining the overlying photoreceptors, mediating the uptake of nutrients, ions, and water, and phagocytizing the shed photoreceptor outer segment [1]. Several retinal degenerative diseases, including age-related macular degeneration (AMD), are closely related with RPE dysfunction [2]. Due to intense illumination from focal light, high oxygen tension in the macular area, and phagocytosis of photoreceptor outer segments, RPE cells are specifically sensitive to oxidative stress [3]. As a result, the RPE is constantly damaged by oxidative stress. Although the exact pathogenesis of AMD remains largely unknown, oxidative stress-induced RPE damage is believed to be one of the primary causes [4,5]. Therefore, cellular oxidative stress plays an important role in RPE cell death during aging and the development of AMD, the primary cause of blindness in elderly persons [6]. Therefore, understanding the mechanisms of RPE cell dysfunction under oxidative stress conditions is critical for developing new therapies for AMD.

Pentraxins are soluble pattern recognition receptors, within a family of proteins that contain a pentraxin domain with pentraxin signature (HxCxS/TWxS) in their carboxy-terminal region [7,8]. Pentraxins are a superfamily of conserved proteins, characterized by a cyclic, multimeric structure and a conserved C-terminal domain. Based on the primary structure, pentraxins are divided into two groups, termed short and long pentraxins. Classic pentraxins, such as C-reactive protein (CRP) and serum amyloid P, are acute phase proteins that are rapidly activated in response to inflammation. Pentraxin 3 (PTX3; also called tumor necrosis factor-alpha [TNF-α]-stimulated gene 14) is the prototypic long pentraxin, which shares similarity with the classic pentraxin in the C-terminal domain but has an unrelated N-terminal sequence. Further, PTX3 is an essential component of the innate immune system [9]. It is rapidly produced and released by several cell types, including RPE cells, such as inflammatory signals, and plays a non-redundant role in controlling inflammation.

Previously, we reported that PTX3 is expressed and secreted in response to either pro-inflammatory mediators, such as IL-1β and TNF-α, or endoplasmic reticulum stress inducer, tunicamycin, in human RPE cells [3]. We described that plasma PTX3 levels were elevated in patients with neovascular AMD [10]. However, the expression and molecular mechanisms of PTX3 in response to NaIO_3_-induced oxidative stress have not been investigated in RPE cells. Here, we demonstrated that NaIO_3_, a known oxidative toxic agent, induced the expression of PTX3, with Akt and reactive oxygen species (ROS) signaling pathways playing a role in the molecular mechanisms. Moreover, oxidative stress-induced PTX3 is involved in the oxidative stress response and the expression of AMD-related genes, including complement factor I (*CFI*), complement factor H (*CFH*), apolipoprotein E (*APOE*), and toll-like receptor 4 (*TLR4*), in human RPE cells and accelerated RPE cell death. Taken together, these results provide critical insights into the pathologic effects of PTX3 during oxidative stress in the early development of age-related macular degeneration.

## 2. Results

### 2.1. NaIO_3_ Treatment Increases mRNA and Protein Levels of PTX3 in Human Retinal Pigment Epithelial Cells

In general, oxidative stress is a well-known stimulus for RPE dysfunction in aging and the development of AMD [6,11,12]. Previously described observations suggest that PTX3 expression was enhanced in response to either pro-inflammatory mediators or endoplasmic reticulum stress inducers in human RPE cells. Specifically, plasma PTX3 levels were elevated in patients with neovascular AMD [10,13]. However, the expression and biological functions of PTX3 under oxidative stress conditions in RPE cells have not yet been studied. We analyzed both mRNA and protein expression levels of PTX3 in H-RPE and ARPE-19 cells in response to oxidative stress inducer, NaIO_3_. We isolated total RNA from human primary H-RPE and ARPE-19 cells after treatment with NaIO_3_ for the indicated doses (50 nM, 100 μM, 500 μM, 1 mM, and 2.5 mM) at 24 h (Figure 1A and Appendix A). Using quantitative real time (RT)-PCR, PTX3 mRNA expression increased in H-RPE (500 μM NaIO_3_ treatment) and ARPE-19 cells (100 μM NaIO_3_ treatment). mRNA levels of PTX3 were 2.23 ± 0.03-fold higher in H-RPE cells and 3.01 ± 0.01-fold higher in ARPE-19 cells in the presence of 500 μM and 100 μM NaIO_3_, respectively, compared with the vehicle treatment. Additionally, PTX3 mRNA levels began to increase and reached maximum expression 24 h after NaIO_3_ treatment in H-RPE (500 μM) and ARPE-19 cells (100 μM) (Figure 1B and Appendix A). The results indicate that NaIO_3_ upregulates the transcriptional level of PTX3 in retinal pigment epithelial cells. We next investigated PTX3 protein levels using ELISA methods after collecting supernatants post-NaIO_3_ administration in H-RPE and ARPE-19 cells. Similar to the data of Figure 1, the protein levels of PTX3 were upregulated with administration of various doses of NaIO_3_ for 48 h in H-RPE (500 μM) and ARPE-19 cells (100 μM) (Figure 2A and Appendix A). The protein levels of PTX3 were markedly increased 24 and 48 h after NaIO_3_ treatment compared with the vehicle treatment in H-RPE (500 μM) and ARPE-19 cells (100 μM) (Figure 2B and Appendix A). These data suggest that NaIO_3_ promoted oxidative stress, which resulted in increased PTX3 mRNA and protein expression in human retinal pigment epithelia cells. 

### 2.2. NaIO_3_-Activated ROS, Akt, and ERK Signaling Pathway Were Regulations of PTX3 Expression in Human Retinal Pigment Epithelial Cells

To identify the signaling molecules involved in regulating PTX3 expression by NaIO_3_, we isolated protein from H-RPE cells at various time points after NaIO_3_ (500 μM) administration. NaIO_3_ did not have a significant effect on overall unphosphorylated Akt, extracellular signal-regulated kinase (ERK), c-jun N-terminal kinase (JNK), p38, and inhibitor of kappa α (IκBα). The phosphorylation and expression of the signaling molecules over time were slightly altered by NaIO_3_ administration, however, phosphorylation of Akt at Thr308 and Ser473, and phosphorylated ERK were increased by NaIO_3_ in H-RPE cells (Figure 3A). Although phosphorylation of ERK was increased, phosphorylation of p38, JNK, and IκBα were weak in response to NaIO_3_. We then assessed which signaling pathway(s) were responsible for stimulating PTX3 production upon NaIO_3_ exposure in H-RPE cells. We used specific inhibitors of LY294002 (PI3 kinase inhibitor), N-acetyl-L-cysteine (NAC, cytosolic ROS scavenger), U0126 (mitogen-activated protein kinase kinase 1/2 inhibitor, MEK1/2 inhibitor), SB203580 (p38 MAP kinase inhibitor), SP600125 (JNK MAP kinase inhibitor), and Bay 11–7082 (NF-κB inhibitor), respectively [14,15,16,17]. The H-RPE cells were treated with LY294002 (5 μM), U0126 (1 μM), SB203580 (10 μM), SP600125 (5 μM), and Bay 11–7082 (1 μM), in the presence or absence of NaIO3, and mRNA or protein levels of PTX3 were assessed 24 h or 48 h after administration. LY294002, U0126, and NAC blocked mRNA and protein levels of PTX3 in response to NaIO_3_ (Figure 3B,C). However, SB203580 (10 μM), SP600125 (5 μM), and Bay 11–7082 (1 μM) exerted no effect on PTX3 expression in the presence of NaIO_3_ (Figure 3B,C). These data suggest that the ROS, Akt, and ERK signaling pathways may play a role in PTX3 production in response to NaIO_3_ in human retinal pigment epithelial cells. 

### 2.3. NaIO_3_-Induced mRNA Levels of Antioxidant Enzymes Were Downregulated in PTX3 shRNA Expressing Retinal Pigment Epithelial Cells

To investigate the effects of PTX3 expression under NaIO_3_-induced oxidative condition, we generated hPTX3 shRNA or control shRNA expressing ARPE-19 cells. To check down regulation of PTX3 expression in hPTX3 shRNA expressing ARPE-19 cells compared with control shRNA expressing ARPE-19 cells, total RNA and supernatants were harvested and NaIO_3_-induced PTX3 mRNA and protein levels were analyzed hPTX3 shRNA or control shRNA expressing ARPE-19 cells. mRNA and protein levels of PTX3 were decreased in hPTX3 shRNA expressing ARPE-19 cells compared with control shRNA expressing ARPE-19 cells (Figure 4A,B). Oxidative stress is well known to induce RPE cell death in AMD [18,19]. To further investigate the critical role of PTX3 in AMD pathogenesis, such as oxidative stress, RPE cell death, and AMD-associated gene expression, we examined mRNA levels of antioxidative enzymes in hPTX3 shRNA expressing ARPE-19 cells compared with control shRNA expressing ARPE-19 cells in response to NaIO_3_. We harvested RNA from control or hPTX3 shRNA expressing ARPE-19 cells 24 h after vehicle or NaIO_3_ treatment. Thereafter, mRNA levels of antioxidant enzymes, such as glucose-6-phosphate dehydrogenase (G6PDH), catalase (CAT), glutathione S-reductase (GSR), glutathione peroxidase 1 (GPX1), superoxide dismutase 1 (SOD1), and superoxide dismutase 2 (SOD2), were analyzed using quantitative real-time RT-PCR (Figure 4C–H). mRNA levels of G6PDH, CAT, and GSR increased in response to NaIO_3_ in control shRNA expressing ARPE-19 cells, but not in hPTX3 shRNA expressing ARPE-19 cells (Figure 4C–E). However, mRNA levels of GPX1, SOD1, and SOD2, did not increase in response to NaIO_3_ in both shRNA expressing ARPE-19 cells (Figure 4F–H). 

### 2.4. NaIO_3_-Induced Cell Death and the AMD-Associated Gene Expression Were Diminished in PTX3 shRNA Expressing Retinal Pigment Epithelial Cells

Cell viability was assessed to determine cellular response to NaIO_3_. The cell viability of ARPE-19 cells decreased by 48.78 ± 2.19% in response to 5 mM NaIO_3_ (Figure 5A). Only NAC (cytosolic ROS scavenger) and LY294002 (PI3 kinase inhibitor) rescued the cell viability of ARPE-19 cells up to 94.29 ± 4.71% and 79.40 ± 2.98%, respectively, in response to 5 mM NaIO_3_ (Figure 5B). To identify the role of PTX3 in NaIO_3_-induced RPE cell death, we verified the viability 48 h after 5 mM NaIO_3_ administration in control or hPTX3 shRNA expressing ARPE-19 cells. While NAC and LY294002 rescued the cell viability in response to NaIO_3_ in control shRNA expressing ARPE-19 cells, these inhibitors had no effects on the cell viability in response to NaIO_3_ in hPTX3 shRNA expressing ARPE-19 cells (Figure 5C). 

Thereafter, the effects of PTX3 expression on AMD-associated gene expression in response to NaIO_3_ using control or hPTX3 shRNA expressing ARPE-19 cells were assessed. mRNA levels of AMD-associated genes, such as complement factor I (CFI), complement factor H (CFH), apolipoprotein E (APOE), and toll-like receptor 4 (TLR4), were enhanced 12 h after NaIO_3_ exposure to control shRNA expressing ARPE-19 cells, but not in hPTX3 shRNA expressing ARPE-19 cells (Figure 6). These results suggest that NaIO_3_-induced PTX3 expression could lead to oxidative stress, cell death, and AMD-associated gene expression in RPE cells. Therefore, PTX3 production might play as a pathologic mediator under oxidative condition. 

## 3. Discussion

AMD is a major cause of legal blindness in the elderly in developed countries. Furthermore, millions of AMD patients lose their sight each year, as there is no effective treatment for dry AMD [20,21]. AMD is associated with several risk factors, and many of them are linked to increased oxidative stress. Oxidative stress is a major factor in retinal pigment epithelium (RPE) cell injury that leads to AMD-related pathological changes [22]. NaIO_3_ is an oxidative toxic agent and its selective RPE cell damage allows it to be used as a reproducible AMD model. Despite several publications using this model to describe cell death and molecular events underlying oxidative stress-induced cellular responses mimicking the pathogenesis of AMD, cell viability control remains unclear in RPE cells. Although the role of PTX3 in many diseases is controversial, PTX3 is considered an inflammatory marker in many inflammatory diseases, including vascular disease. Previously, we demonstrated the expression of PTX3 in response to inflammatory stimuli and ER stress inducers [23]. We reported that plasma PTX3 levels were elevated in patients with neovascular AMD [10]. However, the expression and effects of PTX3 in NaIO_3_-induced signaling pathways and cell viability have not been elucidated in RPE cells. In this study, we showed that NaIO_3_ induces mRNA and protein levels of PTX3 in H-RPE and ARPE-19 cells. The expression and functions of PTX3 have been described in RPE cells, including our previous studies. Our research has indicated the expression and importance of PTX3 in RPE cells. Further, Nissen and colleagues have demonstrated that PTX3 acts as a ligand of complement factor H (CFH), and may participate in AMD immunopathogenesis [24]. Handa and colleagues have shown that PTX3 activity is induced by oxidative stress inducer, 4-hydroxynonenal (4-HNE), and acts as an essential brake for complement and inflammasome activation in ARPE-19 cells [25]. However, we have observed different expression levels of inflammatory cytokines, IL-6, IL-1β, and TNF-α, in the presence of NaIO_3_ in PTX3 shRNA expressing ARPE-19 cells compared with control shRNA expressing ARPE-19 cells (Appendix A). In a recent study, they found that NaIO_3_ can induce cytosolic ROS but not mitochondrial ROS production and activate ERK, p38, JNK, and Akt signaling pathway [26]. Especially, they described that cytosolic ROS-dependent p38 and JNK activation lead to cell death in NaIO_3_-treated ARPE-19 cells. Furthermore, they showed that cytosolic ROS-mediated autophagy and balance of mitochondrial dynamics contribute to cell survival, also. In their study, they suggested that NaIO_3_-induced ROS could simultaneously regulate multiple cellular events. In this study, NaIO_3_ also induced activation of p38, JNK, and EKR signaling pathway in H-RPE cells. In the presence of NaIO_3_, ROS and the phosphorylation of Akt and ERK were involved in PTX3 expression in H-RPE cells. Handa’s group asserted that 4-HNE-induced PTX3 exerts protective effects against oxidative stress-induced complement and inflammasome activation [11]. NaIO_3_-induced RPE cell death was rescued in PTX3 shRNA expressing ARPE-19 cells compared with control shRNA expressing ARPE-19 cells. More importantly, the expression of oxidative stress-induced antioxidant enzymes, G6PDH, CAT, and GSR, and AMD-associated genes, CFI, CFH, APOE, and TLR4, were decreased in PTX3 shRNA expressing ARPE-19 cells. These data suggest that oxidative stress and a risk for AMD were reduced in PTX3 shRNA expressing ARPE-19 cells. In this study, we provide information regarding the critical role of PTX3 under oxidative stress conditions in the early stage of AMD development, especially the loss of RPE cells. 

## 4. Materials and Methods 

### 4.1. Reagents

Sodium Iodate (NaIO_3_; 71702) was purchased from Sigma-Aldrich (St. Louis, MO, USA). Retinal Pigment Epithelial Cell Growth Medium (RtEGM^TM^; #00195407) with supplements including 2% FBS, 2% L-glutamine, 0.5% bFGF, 0.1% GA-1000 was purchased from LONZA (Walkersville, MD, USA). Dulbecco’s modified Eagle’s medium (DMEM; 12800-017), fetal bovine serum (FBS; 26140-079), penicillin/streptomycin (10378-016), and 0.25% trypsin (25200-072) and other cell culture reagents were purchased from Gibco (Gaithersburg, MD, USA). Primary antibodies including phosphor-Akt (ser473; #4058, Thr308; #9275), total Akt (#9272), phospho-ERK (#4370), ERK (#4695), phosphor-JNK (#9251), JNK (#9252), phosphor-p38 (#4511), p38 (#9212), phosphor-IκBα (#9246), and IκBα (#9242) (Cell Signaling Technology, Inc., Danvers, MA, USA), and β-actin polyclonal antibody (SC-47778) (Santa Cruz Biotechnology, Inc., Santa Cruz, CA, USA) were used for western blotting analysis. Human pentraxin 3 (PTX3; DY1826) ELISA kit was purchased from R&D System, Inc. (Minneapolis, MN, USA). U0126 (BML-EI282), LY194002 (BML-ST420), SP600125 (BML-EI305), SB203580 (BML-EI286), BAY 11-7082 (BML-EI278) (Enzo Life Sciences, Farmingdale, NY, USA), and NAC (A7250) (Sigma-Aldrich, St. Louis, MO, USA) were used for inhibitor reagents.

### 4.2. Human Retinal Pigment Epithelial (RPE) Cell Culture

Primary human fetal RPE (H-RPE; #00195406) cells were purchased at passage one from LONZA (Walkersville, MD, USA), and all experiments were performed with cells between passage two to six. The cells were maintained in RtEGM^TM^ medium supplemented with 2% FBS, 2% L-glutamine, 0.5% bFGF, 0.1% GA-1000. Human retinal pigmented epithelial cell lines ARPE-19 cells (CRL-2302TM) were purchased from the American Type Culture Collection (ATCC, Manassas, VA, USA). ARPE-19 cells were cultured in Dulbecco’s modified Eagle’s medium supplemented with 10% FBS, 100 U/mL penicillin and streptomycin. Cell cultures were maintained at 37 °C in a humid atmosphere incubator with 5% CO_2_ and 95% air. The medium was changed every 3–4 days. To passage the cells, we subcultured RPE cells at a 1:4 dilution using 0.25% trypsin and cells usually reached confluence after about four days.

### 4.3. Quantitative Real-Time Reverse Transcription-Polymerase Chain Reaction (qRT-PCR)

For cultured cells, total RNA was isolated from cultured cells using TRIzol reagent (#15596018) (Thermo Fisher Scientific, Inc., Waltham, MA, USA). Equal amounts of RNA were reverse transcribed with SuperScript™ III First-Strand Synthesis System (#18080-044) (Thermo Fisher Scientific, Inc., Waltham, MA, USA) to cDNA. qRT-PCR was performed on the resulting cDNA using iQ SYBR Green Supermix (#170-8882AP) (Bio-Rad Laboratories, Inc., Hercules, CA, USA). The comparative cycle threshold (Ct) value method, representing log transformation, was used to establish relative quantification of the fold changes in gene expression using StepOne plus system (Applied Biosystem, CA, USA). β-actin was used as an internal control (a commonly used loading control for gene degradation in PCR). Primers of human β-actin, pentraxin 3 (PTX3), Glucose-6-phosphate dehydrogenase (G6PDH), Glutathione S-reductase (GSR), Glutathione peroxidase 1 (GPX1), Superoxide dismutase 1 (SOD1), Superoxide dismutase 2 (SOD2), catalase (CAT), complement factor H (CFH), complement factor I (CFI), Apolipoprotein E (APOE), and Toll-like receptor 4 (TLR4) were purchased from Cosmo Genetech, Inc. (Seoul, Korea). The primer pairs which we used were listed in Table 1. Amplification of cDNA started with 10 min at 95 °C, followed by 40 cycles of 15 s at 95 °C and 1 min at 60 °C.

### 4.4. Enzyme-Linked Immunosorbent Assay (ELISA)

Cell culture supernatants were used to measure human PTX3 using ELISA kits from R&D Systems (Minneapolis, MN, USA) according to the manufacturer’s instructions. In brief, the ELISA plates (BD Biosciences, San Jose, CA) were coated with a monoclonal antihuman PTX3 antibody (2 μg/mL) in coating buffer (1% bovine serum albumin in PBS (150 mM NaCl, 5 mM KCl, 5 mM Na_2_HPO_4_, 2 mM KH_2_PO_4_; pH 7.2–7.4) for overnight at room temperature. Then the plates were blocked with coating buffer for 2 h at room temperature, and incubated with either recombinant human PTX3 standards or the samples collected in quadruplicate (100 μL/well) for another 2 h. The plates were then incubated with a biotinylated human PTX3 antibody (150 ng/mL) for 2 h, and freshly diluted streptavidin-horse radish peroxide (HRP) for 20 min subsequently in the dark. After each step, the plates were washed three times with the washing buffer. The chromogen substrate tetra-methylbenzidine (100 μL/well; eBioscience, Inc., San Diego, CA, USA) was added and incubated for 5 min in the dark. The reaction was stopped by adding 2 N H_2_SO_4_ (50 μL/well), and the plates were read at 450 nm with an automatic ELISA reader (MERK SensIdent Scan, Helsinki, Finland).

### 4.5. Western Blot Analysis

The RPE cells were harvested using radioimmunoprecipitation assay (RIPA) buffer (Tris/Cl (pH 7.6); 100 mmole/L, ethylenediaminetetraacetic acid (EDTA); 5 mmole/L, NaCl; 50 mmole/L, β-glycerophosphate; 50 mmole/L, NaF; 50 mmole/L, Na_3_VO_4_; 0.1 mmole/L, NP-40; 0.5%, Sodium deoxycholate; 0.5%) with 1× Complete™ protease inhibitor Cocktail (#39922700) (Roche Applied Science, Mannheim, Germany). Protein concentrations of cell lysates were determined using the Pierce BCA protein assay kit (#23225) (Thermo Scientific, Rockford, IL, USA). The samples were resolved with 12% sodium dodecyl sulfate-polyacrylamide gel electrophoresis (PAGE) gels and transferred to polyvinylidene difluoride (PVDF) membranes (Bio-Rad) overnight (120 mA). Membranes were blocked for 2 h at room temperature with a 5% nonfat milk solution in tris-buffered saline with Tween 20 (TBST) buffer (20 mM Tris–HCl, pH 7.4, 500 mM NaCl, 0.1% Tween 20). The blots were then incubated with various antibodies (diluted 1:1000; Cell Signaling Technology, Inc.) in TBST overnight at room temperature. Equal loading was confirmed with an anti- β-actin (Santa Cruz Biotechnology, Inc., Santa Cruz, CA, USA). The blots were then washed three times in TBST and incubated with an anti-rabbit secondary antibody or an anti-mouse secondary antibody in TBST for 1 h at room temperature. Finally, immunoblots were detected by SuperSignal^®^ West Pico Chemiluminescent Substrate (#34580) (Thermo Fisher Scientific, Inc., Waltham, MA, USA) and visualized after exposure to X-ray film.

### 4.6. Construction of hPTX3 shRNA Expressing ARPE-19 Cells

Human PTX3 shRNA and nonspecific control shRNA (Sigma-Aldrich, St. Louis, MO, USA) were transfected into ARPE-19 cells using transfection reagents (#E2691) (Promega, Madison, WI, USA) according to the protocol of the manufacturer. Briefly, for each transfection, shRNA (1 μg) was added to ARPE-19 cells for 24 h, and stable clones expressing shRNA were further selected by puromycin (1.0 μg/mL). Cell culture medium containing puromycin was renewed every 48 h, until resistant colonies could be identified. The expression of PTX3 and the loading control (β-actin) in stable cells was tested.

### 4.7. Cell Viability Assay

Cell viability was determined by the MTS assay using the Cell Titer 96 AQueous one solution cell proliferation assay kit (G358B) (Promega, Madison, WI, USA). Cells were seeded at 0.7 × 10^4^ cells per well in 96-well plates. After reagent treatment, 20 µL of MTS solution was added to each well, and plates were incubated for an additional 2 to 4 h at 37 °C. Absorbance was then measured at 490 nm using a SpectraMax M2 microplate reader (Molecular Devices, Sunnyvale, CA, USA) to calculate cell survival percentages.

### 4.8. Statistical Analysis

Data represent mean ± SD. For comparisons between two groups, we used the Student’s two-tailed unpaired t test. For comparisons of timed series experiments, we performed Student paired t tests. The Mann-Whitney U test was performed to compare mRNA expression of PTX3 after NaIO_3_ administration and antioxidant enzymes and AMD-associated genes between control shRNA and PTX3 shRNA expressing ARPE-19 cells. Statistically significant differences were accepted at *p* < 0.05.

## 5. Conclusions

NaIO_3_ increased PTX3 expression through PI3 kinase and ERK signaling pathway and cytosolic ROS in human retinal pigment epithelial cells. Human PTX3 shRNA expressing cells were resistant to NaIO_3_-induced cell death. Furthermore, NaIO_3_-enhanced mRNA expression of antioxidant enzymes, such as G6PDH, CAT, and GSR and AMD marker genes, such as CFI, CFH, APOE, and TLR4 in hPTX3 shRNA expressing RPE cells. These results suggest that PTX3 accelerates RPE cell death and might be involved in AMD development in the presence of oxidative stress.

## Figures and Tables

**Figure 1 ijms-20-06028-f001:**
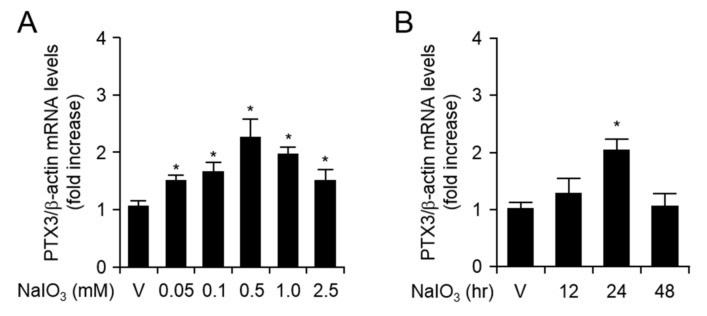
The expression of pentraxin 3 (PTX3) mRNA levels was enhanced after NaIO_3_ administration in human RPE cells. Primary human H-RPE cells were treated for 24 hours in various doses of NaIO_3_ (**A**). H-RPE cells were exposed to 500 μM, for the indicated time points (**B**). Second and third of the H-RPE cells were used. Values are presented as mean ± SD, n = 3. * *p* < 0.05, increased PTX3 mRNA expression after NaIO_3_ administration vs vehicle (V).

**Figure 2 ijms-20-06028-f002:**
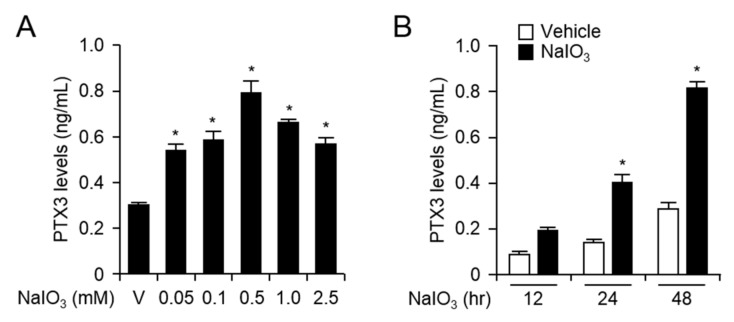
The protein levels of PTX3 were enhanced after NaIO_3_ administration in human RPE cells. Primary human H-RPE cells were treated for 48 hours in various doses of NaIO_3_ (**A**). H-RPE cells were exposed to 500 μM, for the indicated time points supernatants were harvested and analyzed for PTX3 production (**B**). Third and fourth passages of the H-RPE cells were used. Values are presented as mean ± SD, n = 12. * *p* < 0.05, increased PTX3 after NaIO_3_ administration vs vehicle (V).

**Figure 3 ijms-20-06028-f003:**
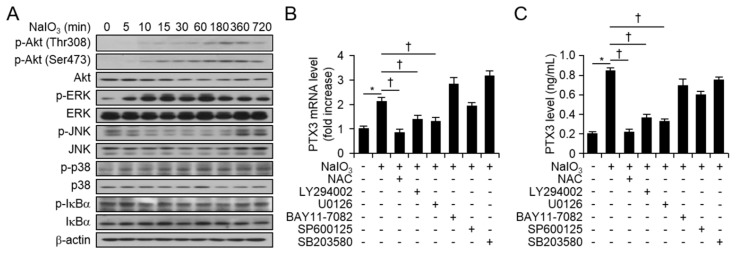
ROS and PI3 kinase signaling pathways are involved in PTX3 induction by NaIO_3_ in human RPE cells. The levels of Akt, phosphorylated Akt (Ser473 and Thr308), total ERK, phosphorylated ERK, total JNK, phosphorylated JNK, total IκBα, phosphorylated IκBα, total p38, and phosphorylated p38 proteins were assessed using western blotting analysis (**A**). β-actin was used as a loading control. Experiments were performed at least three independent times. Total RNA was extracted from H-RPE cells 24 h after 500 µM NaIO_3_ with signaling inhibitor (1 µM BAY11-7082, 1 µM U0126, 10 µM SB203580, 5 µM SP600125, 5 µM LY2940002, or 5 mM NAC), administration. Quantitative real-time RT-PCR was performed to assess mRNA levels of *PTX3*. For all real-time PCR analyses, mouse β-actin was used as a control for normalization. Expression levels of each mRNA are divided by expression of β-actin and shown as a ratio of each mRNA/β-actin. Values are presented as mean ± SD, *n* = 3 (**B**). Supernatants were harvested from H-RPE cells 48 h after NaIO_3_ administration with signaling inhibitors (**C**). Supernatants were harvested and measured for PTX3 production using human PTX3 ELISA kit. Third and fifth passages of the H-RPE cells were used. Values are presented as mean ± SD, *n* = 12. **p <* 0.05, increased PTX3 after NaIO_3_ administration *vs* vehicle (V). †*p* < 0.05, decreased PTX3 in response of NaIO_3_ plus signaling inhibitor *vs.* NaIO_3_ alone.

**Figure 4 ijms-20-06028-f004:**
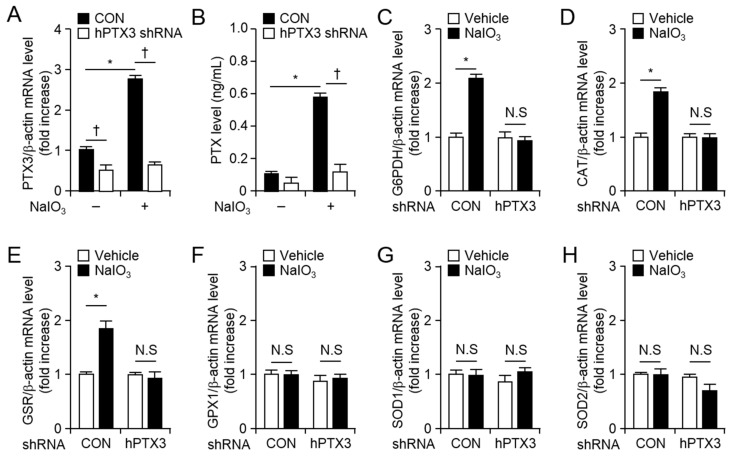
mRNA levels of antioxidant enzymes were decreased in hPTX3 shRNA expressing ARPE-19 cells in response to NaIO_3_. Total RNA was extracted from control or PTX3 shRNA expressing ARPE-19 cells 24 h after 100 µM NaIO_3_. mRNA expression (**A**) and protein levels (**B**) of PTX3 were analyzed. mRNA levels of *G6PDH* (**C**), *CAT* (**D**), *GSR* (**E**), *GPX* (**F**), *SOD1* (**G**), and *SOD2* (**H**) were analyzed by quantitative real-time RT-PCR. Human β-actin was used as a control for normalization. Expression levels of each mRNA are divided by expression of β-actin and shown as a ratio of each mRNA/β-actin. Fifth and sixth passages of the H-RPE cells were used. Values are presented as mean ± SD, *n* = 3. †*p* < 0.05, decreased mRNA levels of antioxidant enzyme after NaIO_3_ administration *vs* vehicle. **p <* 0.05, increased mRNA levels of antioxidant enzyme after NaIO_3_ administration *vs* vehicle. N.S. indicates non-significance.

**Figure 5 ijms-20-06028-f005:**
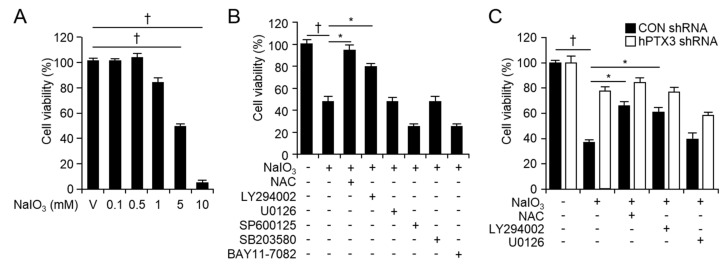
Cell viability was enhanced in hPTX3 shRNA expressing ARPE-19 cells in response to NaIO_3_. The cell viability was analyzed 48 h after various doses of NaIO_3_ administration in ARPE-19 cells (**A**). †*p* < 0.05, decreased the cell viability after NaIO_3_ administration *vs* vehicle (V). The cell viability was analyzed in response to NaIO_3_ or NaIO_3_ plus signaling inhibitors in ARPE-19 cells (**B**). †*p* < 0.05, decreased the cell viability after NaIO_3_ administration *vs* vehicle (V). **p <* 0.05, rescued the cell viability in response to NaIO_3_ plus signaling inhibitors *vs* NaIO_3_ alone. The cell viability was analyzed in response to NaIO_3_ or NaIO_3_ plus signaling inhibitors in control or hPTX3 shRNA expressing ARPE-19 cells (**C**). Fourth and fifth passages of the H-RPE cells were used. †*p* < 0.05, decreased the cell viability after NaIO_3_ administration *vs* vehicle (V). **p <* 0.05, rescued the cell viability in response to NaIO_3_ plus signaling inhibitors *vs* NaIO_3_ alone. Values are presented as mean ± SD, *n* = 12.

**Figure 6 ijms-20-06028-f006:**
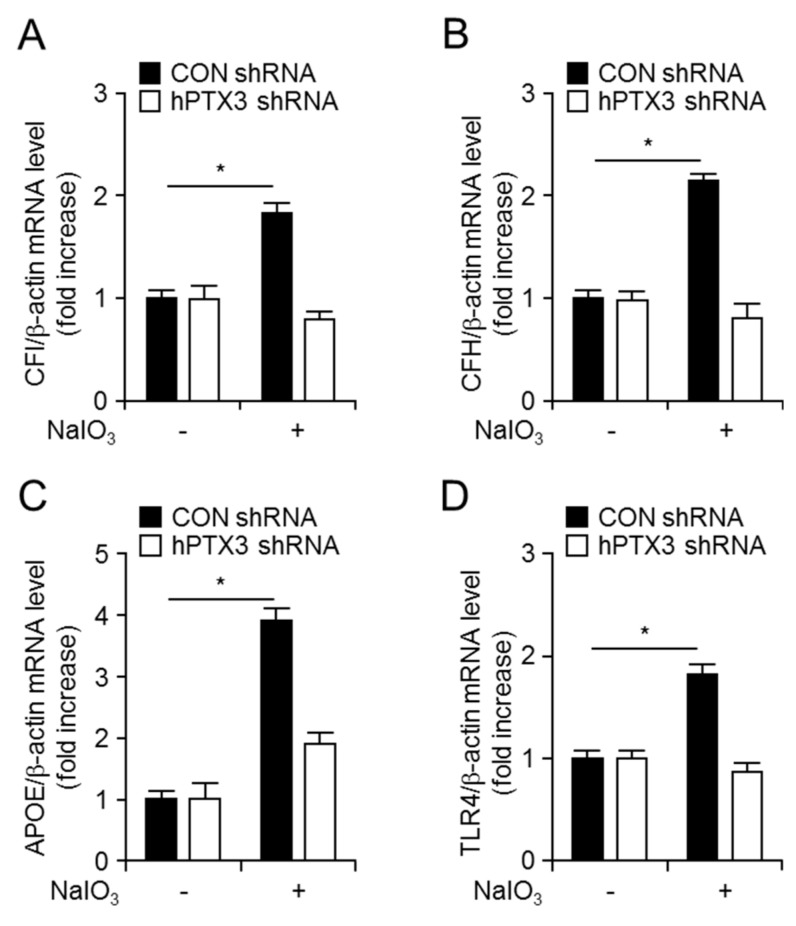
mRNA levels of AMD-associated genes were suppressed in hPTX3 shRNA expressing ARPE-19 cells in response to NaIO_3_. Total RNA was extracted from control or hPTX3 shRNA expressing ARPE-19 cells 12 h after 100 µM NaIO_3_ administration. mRNA levels of *CFI* (**A**), *CFH* (**B**), *APOE* (**C**), and *TLR4* (**D**) were analyzed by quantitative real-time RT-PCR. Human β-actin was used as a control for normalization. Expression levels of each mRNA are divided by expression of β-actin and shown as a ratio of each mRNA/β-actin. Fourth passage of the H-RPE cells was used. Values are presented as mean ± SD, *n* = 3. **p <* 0.05, increased mRNA levels of genes after NaIO_3_ administration *vs* vehicle.

**Table 1 ijms-20-06028-t001:** The primers sequences were as follows.

Human Gene	Forward Primer Sequence 5′ to 3′	Reverse Primer Sequence 5′ to 3′
PTX3	AATGCATCTCCTTGCGATTC	TGAAGTGCTTGTCCCATTCC
G6PDH	TGAGCCAGATAGGCTGGAA	TAACGCAGGCGATGTTGTC
GSR	TCACCAAGTCCCATATAGAAATC	GTGTAGGACTAGCGGTGT
GPX1	CCAAGCTCATCACCTGGTCT	TCGATGTCAATGGTCTGGAA
SOD1	GAAGGTGTGGGGAAGCATTA	ACATTGCCCAAGTCTCCAAC
SOD2	CGTGACTTTGGTTCCTTTGAC	AGTGTCCCCGTTCCTTATTGA
CAT	CGTGCTGAATGAGGAACAGA	AGTCAGGGTGGACCTCAGTG
CFH	TACTGGCTGGATACCTGCTC	CCTGACGGAGTCTCAAAATG
CFI	GGTGAGGTGGACTGCATTACA	CCTCCCACAATTCGTTTCCTTC
APOE	AACTGGCACTGGGTCGCTTT	GCCTTCAACTCCTTCATGGTCTCGT
TLR4	ACTTGGACCTTTCCAGCAAC	TTTAAATGCACCTGGTTGGA
β-actin	ATCGTGCGTGACATTAAGGAGAAG	AGGAAGGAAGGCTGGAAGAGTG

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
