# Peer review of "Oxidative Stress-Induced Pentraxin 3 Expression Human Retinal Pigment Epithelial Cells Is Involved in the Pathogenesis of Age-Related Macular Degeneration"

_ijms, 2019, doi:10.3390/ijms20236028_

Round 1

Reviewer 1 Report

The manuscript "Oxidative stress-induced pentraxin 3 expression is involved in the pathogenesis of age-related macular degeneration in human retinal pigment epithelial cells" by Hwang et al presents an interesting study that shows the involvement of sodium iodate-induced Pentraxin 3 in oxidative stress in RPE cells in vitro. Revision as per the below comments is recommended. Comments: 1. Introduction: line 75 – should probably be ‘RPE’ instead of PRE. 2. Results: • 2.1.: line 80 – ‘in’ aging instead of ‘during’ aging. • Line 88 - Please mention the ‘indicated doses’. • 2.2.: line 115 – involved in ‘regulation of’PTX3 expression. • Please provide all Mean±SD values in the results section. • Figure 4 legend – Mention/explain the sign used to represent p values. • Please describe vehicle treatment (concentration, etc.) used in this study. 3. Methods: • Please provide the catalog numbers of sodium iodate, all antibodies, ELISA kits, RIPA buffer, and other kits and reagents used in this study. • Since the sample size is small, non-parametric Mann-Whitney tests would be more appropriate for statistical analysis of results. • Authors mention that cell culture passage numbers 2-6 were used in this study (line 267). Please explain the exact passage numbers used for each experiment and the reason for using a different passage number for each experiment. • Provide primer sequences in table format so it is easier to follow. 4. Discussion: This section reads like a results section and needs a lot of work! • Get rid of figure numbers. • To elaborate this section, please compare and contrast the model used and the findings of this study with previous literature. • Discuss the relevance and limitations of the current study in detail. • In the last paragraph, a brief summary followed by future directions is recommended.

Author Response

We thank the Reviewers for their detailed and helpful comments. We have revised the manuscript accordingly as described in the point-by-point response below. To this end, we have performed the additional experiments suggested by the reviewers. We hope that that these changes will render the manuscript suitable for publication in International Journal of Molecular Sciences. We have extensively addressed all the other concerns of the Reviewers as detailed below.

Reviewers’ Comments:

Reviewer 1

The manuscript "Oxidative stress-induced pentraxin 3 expression is involved in the pathogenesis of age-related macular degeneration in human retinal pigment epithelial cells" by Hwang et al presents an interesting study that shows the involvement of sodium iodate-induced Pentraxin 3 in oxidative stress in RPE cells in vitro. Revision as per the below comments is recommended.

Introduction: line 75 – should probably be ‘RPE’ instead of PRE.

We appreciate the comments by the Reviewer 1 and ‘PRE’ has been corrected in line 75. It now reads ‘RPE’.

Results: 2.1.: line 80 – ‘in’ aging instead of ‘during’ aging. Line 88 - Please mention the ‘indicated doses’. 2.2.: line 115 – involved in ‘regulation of’PTX3 expression. Please provide all Mean±SD values in the results section. Figure 4 legend – Mention/explain the sign used to represent p values. Please describe vehicle treatment (concentration, etc.) used in this study.

All comment of the Reviewer 1 regarding Results had been corrected and added. Please, check blue colors in Results and Materials and Methods sections.

Methods: Please provide the catalog numbers of sodium iodate, all antibodies, ELISA kits, RIPA buffer, and other kits and reagents used in this study. Since the sample size is small, non-parametric Mann-Whitney tests would be more appropriate for statistical analysis of results. Authors mention that cell culture passage numbers 2-6 were used in this study (line 267). Please explain the exact passage numbers used for each experiment and the reason for using a different passage number for each experiment. Provide primer sequences in table format so it is easier to follow.

All comment of the Reviewer 1 regarding Methods had been added and modified. Please, check blue colors in Materials and methods section.

Discussion: This section reads like a results section and needs a lot of work! Get rid of figure numbers. To elaborate this section, please compare and contrast the model used and the findings of this study with previous literature. Discuss the relevance and limitations of the current study in detail. In the last paragraph, a brief summary followed by future directions is recommended.

Discussion has been modified by the Reviewer 1’s comments.

Reviewer 2 Report

There is a very relevant study reported this year by Chan et al in J Biomed Sci. that was not acknowledged at all in this manuscript. 

A part of the results where the authors say in the abstract "We found phosphorylated AKT, a downstream target of the PI3 kinase pathway, phosphor-ERK, and intracellular reactive oxygen species (ROS) were predominantly induced by NaIO3" has already shown by the other group whos paper was not even referenced. 

I would strongly advise the authors to take this into account in their manuscript accordingly before they re-submit the revised paper.

Author Response

We thank the Reviewers for their detailed and helpful comments. We have revised the manuscript accordingly as described in the point-by-point response below. To this end, we have performed the additional experiments suggested by the reviewers. We hope that that these changes will render the manuscript suitable for publication in International Journal of Molecular Sciences. We have extensively addressed all the other concerns of the Reviewers as detailed below.

Reviewer 2

There is a very relevant study reported this year by Chan et al in J Biomed Sci. that was not acknowledged at all in this manuscript. 

A part of the results where the authors say in the abstract "We found phosphorylated AKT, a downstream target of the PI3 kinase pathway, phosphor-ERK, and intracellular reactive oxygen species (ROS) were predominantly induced by NaIO3" has already shown by the other group whos paper was not even referenced. 

I would strongly advise the authors to take this into account in their manuscript accordingly before they re-submit the revised paper.

We appreciate the Reviewer 2 for the helpful comments. The publication from Chan’s group had been addressed in Discussion and added in References, 26.

Reviewer 3 Report

Age-related macular degeneration is a broad term that refers to both wet and dry type diseases. Sodium iodate is a potent oxidizing agent that induces RPE loss, which mimics geographic atrophy. It would be better to specify this point. As the clinical evidence of the involvement of pentraxin 3 in the development of AMD, they provided their own reference which showed elevated serum pentraxin 3 levels in wet ADM patients, which does not correlate with this paper. While RPE dysfunction has a major role in the wet AMD, it is still different from geographic atrophy.

Figure 2. The authors measured pentraxin 3 protein levels from supernatants of RPE cell culture. The hypothesis is that the expression of pentraxin 3 induced by sodium iodate induced RPE death. It would be better to measure pentraxin 3 levels from RPE cells, not supernatants. 

Page 4, Line 125. Please provide the references of those inhibitors. Each inhibitor may show different actions according to its doses. 

Figure 3. It seems like that the expression of p-AKT (Thr308) was not increased significantly. It would be better to quantify the protein expression.

Figure 4 and 5. Please confirm the expression of those molecules in protein levels. 

Page 6, Line 181. Please provide detailed method for the measurement of cell viability. It looks like these results have the most significant meaning for clinical correlation. 

Author Response

We thank the Reviewers for their detailed and helpful comments. We have revised the manuscript accordingly as described in the point-by-point response below. To this end, we have performed the additional experiments suggested by the reviewers. We hope that that these changes will render the manuscript suitable for publication in International Journal of Molecular Sciences. We have extensively addressed all the other concerns of the Reviewers as detailed below.

Reviewer 3

Age-related macular degeneration is a broad term that refers to both wet and dry type diseases. Sodium iodate is a potent oxidizing agent that induces RPE loss, which mimics geographic atrophy. It would be better to specify this point. As the clinical evidence of the involvement of pentraxin 3 in the development of AMD, they provided their own reference which showed elevated serum pentraxin 3 levels in wet ADM patients, which does not correlate with this paper. While RPE dysfunction has a major role in the wet AMD, it is still different from geographic atrophy.

Figure 2. The authors measured pentraxin 3 protein levels from supernatants of RPE cell culture. The hypothesis is that the expression of pentraxin 3 induced by sodium iodate induced RPE death. It would be better to measure pentraxin 3 levels from RPE cells, not supernatants. 

We thank the Reviewer 3 for the strong supportive comments.

PTX3 is a secretory protein when the cells are under stimulation. So measuring PTX3 production from supernatants is more accurate measurement. Also, other publication had been used the same ELISA method to measure PTX3 production from supernatants. Here are some supportive references for measuring PTX3 using ELISA.

References

Bonavita, E.; Gentile, S.; Rubino, M.; Maina, V.; Papait, R.; Kunderfranco, P.; Greco, C.; Feruglio, F.; Molgora, M.; Laface, I.; Tartari, S.; Doni, A.; Pasqualini, F.; Barbati, E.; Basso, G.; Galdiero, MR.; Nebuloni, M.; Roncalli, M.; Colombo, P.; Laghi, L.; Lambris, JD.; Jaillon, S.; Garlanda, C.;, Mantovani, A. PTX3 is an extrinsic oncosuppressor regulating complement-dependent inflammation in cancer. Cell 2015, 160, 700-714. Han, B.; Mura, M.; Andrade, CF.; Okutani, D.; Lodyga, M.; dos, Santos, CC.; Keshavjee, S.; Matthay, M.; Liu, M. TNFalpha-induced long pentraxin PTX3 expression in human lung epithelial cells via JNK. J Immunol 2005, 175, 8303-8311. Norata, GD.; Marchesi, P.; Pirillo, A.; Uboldi, P.; Chiesa, G.; Maina, V.; Garlanda, C.; Mantovani, A.; Catapano, AL. Long pentraxin 3, a key component of innate immunity, is modulated by high-density lipoproteins in endothelial cells. Arterioscler Thromb Vasc Biol 2008, 28, 925-931.

Page 4, Line 125. Please provide the references of those inhibitors. Each inhibitor may show different actions according to its doses. 

Regarding of the references of those inhibitors in line 125, we provided the new references, 14-17.

Figure 3. It seems like that the expression of p-AKT (Thr308) was not increased significantly. It would be better to quantify the protein expression.

We thank the Reviewer 3 for the helpful comments. We did other experiments for Naio3-induced p-AKT expression by western blot analysis and the results had been replaced in Figure 3A.

Figure 4 and 5. Please confirm the expression of those molecules in protein levels. 

We appreciate the Reviewer’s suggestion, but all the antibodies were not available in our lab now. So, we could not finish this experiment within 6 days for revision.

Page 6, Line 181. Please provide detailed method for the measurement of cell viability. It looks like these results have the most significant meaning for clinical correlation. 

We appreciate the Reviewer 3’s comments and we added the detail method for measuring cell viability in the Materials and Methods section.

Round 2

Reviewer 3 Report

I think the authors responded well to the queries. 

Thank you.

Author Response

Reviewer 3

We appreciate that the Reviewer 3 was satisfied with the our response to the Reviewer’ comments and suggestions.

Thanks for your kindness.